# Sleep Disturbances and Mental Well-Being of Preschool Children during the COVID-19 Pandemic in Mexico

**DOI:** 10.3390/ijerph20054386

**Published:** 2023-03-01

**Authors:** Daniela León Rojas, Fabiola Castorena Torres, Salomon Alvarado Ramos, Alfredo del Castillo Morales, Julieta Rodríguez-de-Ita

**Affiliations:** 1Tecnologico de Monterrey, Escuela de Medicina y Ciencias de la Salud, Ave. Morones Prieto 3000, Monterrey 64710, NL, Mexico; 2Tecnologico de Monterrey, Escuela de Medicina y Ciencias de la Salud, Hospital San José, TecSalud, Monterrey 64710, NL, Mexico

**Keywords:** sleep, COVID-19, children, mental health

## Abstract

COVID-19 pandemic confinement caused changes in families and children’s routines worldwide. Studies conducted at the beginning of the pandemic have examined the harmful effects of these changes on mental health, including sleep disturbances. As sleep is essential for optimal childhood development, this study was designed to determine preschool-aged (3–6 years old) children’s sleep parameters and mental well-being during the COVID-19 pandemic in Mexico. Using a cross-sectional design, a survey was applied to parents of preschool children, inquiring about their children’s confinement status, routine changes, and electronics use. The parents responded to the Children’s Sleep Habits Questionnaire and the Strengths and Difficulties Questionnaire to assess children’s sleep and mental well-being. To provide objective sleep data, the children wore wrist actigraphy for seven days. Fifty-one participants completed the assessment. The children’s mean age was 5.2 years, and the prevalence of sleep disturbances was 68.6%. The use of electronic tablets in the bedroom near bedtime and symptoms of mental health deterioration (i.e., emotional distress and behavioral difficulties) were associated with sleep disturbances and their severity. The COVID-19 pandemic’s confinement-related routine changes greatly impacted preschool children’s sleep and well-being. We recommend establishing age-tailored interventions to manage children at higher risk.

## 1. Introduction

COVID-19 pandemic confinement drastically impacted the routines and activities of families worldwide. In Mexico, confinement measures and social distancing were implemented in March 2020. Children were especially affected by the applied restrictions facing school closure, not being allowed in grocery stores and into recreational public spaces. For children, the return to face-to-face school attendance was announced in August 2021, beginning with the regions of the country with the lowest risk of COVID-19 infection spread. By November 2021, most schools had partially returned to face-to-face attendance, and lockdown measures were progressively withdrawn over the following months [1]. Nevertheless, the duration of the implementation of these policies led the population to be exposed to confinement for at least one continuous year.

Emerging studies through the pandemic that evaluated the effects of confinement measures due to the COVID-19 pandemic on mental health found harmful effects on adults’ and children’s mental well-being, reporting an increase in anxiety, depression, and stress symptoms [2,3,4]. In addition, higher rates of sleep disturbances have also been some of the most frequent complaints in adult populations [5]. 

Children’s sleep is regulated by biological, social, parental, and cultural factors, which include routines, daily activity, electronics use, and nutrition. Confinement due to the COVID-19 pandemic has been associated with disturbances in all aforementioned factors, such as more electronics use [6], less physical activity [7], more parental stress [8], and later bedtimes [9]. 

There are mixed data regarding COVID-19 confinement’s impact on preschoolers. Early in the pandemic, Di Giorgio et al. found no significant changes in the proportion of children with sleep difficulties during the lockdown compared to previous data (44.72% vs. 41.6%) but described a general worsening in sleep quality in both children and their mothers [10]. Dellagiulia et al. reported an initial deterioration followed by stabilization of sleep parameters within the first four weeks of confinement in Italy [11]. 

Lecuelle et al. found a significant increase in sleep disorders in young French children, with an increase in the overall score of the Sleep Disorder Scale for Children, a reduced number and duration of naps, and an increased length of nocturnal sleep, rising to 10.9 h in comparison to 10.3 h in a pre-pandemic assessment with no impact on the total sleep time duration over 24 h [12].

Later during the pandemic, Anders et al. described that insomnia in young children was strongly associated with mothers’ reaction to the confinement. A maternal reaction of fear and anxiety could, in turn, expose their children to fewer outings and increase screen viewing. They also described the impact of the lockdown conditions severity to which the population was exposed and the pandemics’ financial impact on the family [13]. Additionally, Mackenzie et al., in a study made in Canada, reported that 40% of children that already had sleep disturbances experienced worsening in their sleep five months after the onset of the pandemic. Parents attributed the worsening of their children’s sleep to a decrease in activity and an increase in electronic device use [14].

Remarkably, most studies during the pandemic assessed children’s sleep patterns through subjective measures such as questionnaires and sleep logs. Although highly standardized questionnaires provide clinically relevant sleep indicators, previous research has suggested divergent results between self-report, parental report, and objective sleep measurements such as actigraphy or polysomnography [15] and could represent a study limitation [16]. 

As actigraphy monitoring allows obtaining sleep data in children’s natural home environment in a non-invasive way and respecting social distancing, it can represent a proper alternative to assess children’s sleep for several consecutive days during the COVID-19 pandemic. Furthermore, objective actigraphic data can provide helpful information that can complement parental reports and bring valuable clinical data [17].

After the prolonged and restrictive confinement measures due to the COVID-19 pandemic in Mexico, there is a growing need to enhance our knowledge about current sleep parameters and mental well-being in early childhood. Preschool children are in a crucial developmental stage to establish healthy bedtime routines [18]. Because of the confinement, they have been exposed to unprecedented routines, activities, and electronic device use changes which could lead to the presentation of sleep disturbances. There is also a consistent evidence base suggesting how sleep disturbances in childhood are associated with several harmful health outcomes, including mental health symptoms (e.g., anxiety, depression, attention deficit hyperactivity disorder, and behavioral problems) and obesity [19,20].

A comprehensive study of sleep parameters and mental well-being of preschool children in Mexico would extensively allow us to identify specific subpopulations with diverse characteristics, offering the possibility of generating prevention strategies in the presentation of actual and future clinical problems, as well as interventions of greater specialization and individualization. Additionally, objective sleep data measures, such as actigraphy, can provide a broader picture of the clinical setting and give valuable insight into future recommendations regarding pandemics. Therefore, this study was designed to describe preschoolers’ mental well-being and sleep parameters by obtaining objective sleep data through actigraphic assessment during the COVID-19 pandemic in Mexico. 

## 2. Materials and Methods

We conducted a cross-sectional study during the COVID-19 pandemic in Mexico from September 2021 to December 2021. After obtaining consent from the group managers, parents and children were invited to participate in this study through two active WhatsApp school groups. Parents were eligible for this study if they were 18 years or older, lived in Mexico during the COVID-19 pandemic, and were the primary caregiver to a preschool-aged child (3–6 years). Children were eligible for this study if they were 3 to 6 years old, lived in Mexico during the COVID-19 pandemic, were not taking any sleep-aid medication, and were not hospitalized or had any infectious disease over the last 12 weeks. The exclusion criterion was an incomplete evaluation. The total sample of this study included 51 predominantly middle-class Mexican parents and children. The mean age of the parents was 36.8 years old; 84% were women, 96% were married, 54.5% had a college degree, and 68% were employed. The children’s mean age was 5.2 years old, and 56% were female.

### 2.1. Clinical Assessment 

Socio-demographic information and the confinement status of the parents and children were obtained using a questionnaire (available as Appendix A). The items included questions about the parents’ occupations and children’s routines, schooling, and electronic use. In addition, the children’s mental well-being was assessed using the Strength and Difficulties Questionnaire (SDQ). Finally, the children’s sleep was subjectively evaluated using the Children’s Sleep Habits Questionnaire (CSHQ) and objectively using wrist actigraphy monitoring for seven consecutive days. 

#### 2.1.1. Rating Scales 

The SDQ is a parental report screening tool with 25 items, whose objective is to assess the mental health status of individuals in the age range of 2–17 years. The 25 items are divided into five scales: emotional symptoms (5 items), behavioral difficulties (5 items), hyperactivity/inattention (5 items), peer relationship problems (5 items), and prosocial behavior (5 items). The total difficulty score is based on 20 items, excluding the prosocial behavior items. The score ranges from 0 to 40 and can be interpreted as a continuous variable or categorized into four groups: 0–13 close to average, 14–16 slightly high, 17–19 high, and 20–40 very high. The SDQ can also provide an externalizing and internalizing symptom score that ranges from 0 to 20 and is the sum of the Behavior and Hyperactivity scales and the Emotional Symptom and Peer Problem scales [21].

The CSHQ is a 45-item parent-rated questionnaire designed to examine sleep behavior in children. The CSHQ includes 33 scored questions rated on a 3-point Likert scale according to the frequency of presentation. The items combined form eight subscales (i.e., bedtime resistance, sleep onset delay, sleep duration, sleep anxiety, night wakings, parasomnias, sleep-disordered breathing, and daytime sleepiness) that create a total score. The total score ranges from 33 to 99, with higher scores representing more sleep disturbances. A score of more than 41 points indicates a pediatric sleep disorder [22].

#### 2.1.2. Actigraphy 

The children’s sleep time was objectively measured using an actigraphy device (GENEActiv Original, Activinsights). Actigraphy is a non-invasive procedure to examine sleep time in an individual’s routine environment. It has been previously validated as a reliable measure of sleep for preschool-aged children [23]. The parents were instructed to place the actigraphy device (preprogrammed to record data at 1 min epoch intervals) on their children’s non-dominant wrist for up to seven consecutive days and nights. This procedure generated the following measures of sleep: total sleep time, total awake time, motionless sleep, light activity during sleep, moderate activity during sleep, and nocturnal awakening. The parents also completed a sleep diary for their children to record hours of bedtime, sleep onset, waking times, overnight events, and times when the device was removed, generating the following measures: total sleep time, sleep latency, and sleep efficiency. The data provided by the diary were used to cross-check the data generated by the actigraphy device.

### 2.2. Data Analyses

The analysis was performed by integrating data from the following sources: The clinical information from the Children’s Sleep Habits Questionnaire (CSHQ), the Strengths and Difficulties Questionnaire (SDQ), a questionnaire on socio-demographic characteristics and electronics use, the sleep diary, and the actigraphy (GENEActiv) devices. 

The 7-day analysis logs from the actigraphy devices were USB downloaded raw in a *.BIN format using the GENEActiv PC Software program. Subsequently, the code utilities from the GENEActiv R Markdown Analysis Tools were used to convert the raw files into a *.CVS format. Detailed reports for each patient file were made. Furthermore, we combine the cases’ information in a single database with case Id, measurement date and time, the movement vector figures (up/down, median, average, and variance), temperature, and magnitude. With the assistance of the GENEActiv R functions, individual estimations of the total sleep time, total awake time, and motion intensity during the day or night were estimated. Local polynomial regression (LOESS) was used to detect nocturnal awakenings and other motion anomalies. Then, the cohort’s figures (average, median, standard deviation, and interquartile range) were then matched with the respective patient ID to analyze the sleep profile with the clinical data.

To obtain detailed group characterization, the research team divided the children into groups—those without sleep disturbances (CSH < 41), those with sleep disturbances (CSHQ > 41), and those with severe (CSHQ > 52) sleep disturbances— and compared each category. The analyses used Fisher’s exact test or χ^2^ for the categorical variables. *t*-tests or Mann–Whitney’s U test were used for quantitative variables. Finally, the children’s sleep time was analyzed using Cox regression to evaluate the sleep alteration hazard risk by the clinical setting of the children. All the statistical analyses were performed using R, version 4.2. 

## 3. Results

A total of 51 parents and children were recruited for this study. Regarding the parents, we found that 84% were women, with a mean age of 36.8 years. Moreover, 96% were married, 65.9% reported having at least two children, 54.5% had a college degree, and 68% were employed. The mean age of the children was 5.27 years, of whom 13.79% had the antecedent of being preterm, and this antecedent was related to higher sleep disturbance severity scores (*p* = 0.04; odds ratio (OR) 6; confidence interval (CI) 95% 1.12–32.24). Furthermore, 56.8% of the children were female, 62% were back to school in partial attendance (i.e., online platform schooling combined with face-to-face attendance), and the rest were in total face-to-face attendance. None of them had a previous diagnosis of mental health disorders. Regarding the use of electronics during the day, the parents reported that 72% of their children had increased their electronic use, with tablets being the most used electronic device, followed by TV and smartphone. The children’s demographics and electronic use during the day are shown in Table 1.

Based on their CSHQ scores, 68.6% of the children experienced sleep disturbances. Of those with sleep disturbances, 23.53% were cataloged in the severe sleep disturbance group with a CSHQ score above 52 points. Bedtime resistance and daytime sleepiness were the two subscales with a more considerable difference in severity between the groups (*p* < 0.01). According to the parental report sleep diary, an average of 9.8 h of total sleep time was found in the entire sample vs. 7 h of total sleep time according to the actigraphy data. As gathered by the sleep diaries, total sleep time and efficiency were found to be related to the CSHQ > 41 score group (*p* = 0.03, *p* = 0.04). Table 2 provides more information on children’s sleep variables and differences between severity groups. 

Regardless of group severity, allowing the use of a tablet in the bedroom near bedtime was associated with sleep disturbances in children (*p* = 0.01; OR = 6.67; 95% CI 1.62 to 27.38), as shown in Table 3. 

Overall, 15.6% of the sample reported an SDQ total score above 14, with 50% ranging slightly high and 50% ranging high. Hyperactivity had the highest score, among all symptoms, followed by behavioral difficulties and emotional distress. Additionally, mental health deterioration symptoms as evaluated using the SDQ total score (*p* < 0.01), externalizing (*p* = 0.01), internalizing (*p* < 0.01), emotional distress (*p* < 0.01), and behavioral difficulties (*p* < 0.01) scores were found to be associated with higher sleep disturbance score. Table 4 provides detailed information on children’s sleep disturbances and mental-well being outcomes. 

Cox regression was used to evaluate the total sleep time and motionless sleep time (actigraphy values) decrease hazard risk. It showed that tablet use in the bedroom near bedtime had statistically significant (*p* = 0.01) influence on both parameters (Figure 1 and Figure 2). 

## 4. Discussion

The COVID-19 pandemic and its associated confinement brought abrupt activity and routine changes worldwide, which generated concerns for children’s mental well-being and sleep. Our data support that sleep disturbances in Mexican preschool-aged children were widespread after 1 year of home confinement during the COVID-19 pandemic (68.6%). These findings are consistent with the ones reported by Lecuelle et al. in France (62%) after the end of their two months of strict confinement [12] and represent higher values of sleep disturbance compared with other international reports of preschool-aged children, such as the one by Wearick-Silva et al. in Brazil during the seventh week of confinement, where they found sleep disturbances in 58.6% of children between 0 and 3 years old and disorders in initiating and maintaining sleep in 33.9% of children between 4 and 12 years old [24]. 

The latter agrees with our findings where bedtime resistance and daytime sleepiness were the most severely affected sleep areas and with Bruni et al. findings in Italy, where a significant delay in bedtime was found, along with an increase (1.9% vs. 5.9%) in daytime sleepiness in children between 4 and 5 years old during COVID-19 pandemic [25]. 

Within our sample, a total sleep time of 9.8 h by parental report vs. 7 h via actigraphy was obtained. As previously mentioned, studies have reported discrepancies over objective vs. subjective sleep measurements in community samples [26], stating that parents tend to overestimate sleep duration (which may be related to the child’s increasing self-regulation capacity during night-wakings). In addition, actigraphy can underestimate sleep time with the presence of motor activity; therefore, complementary use of both instruments can provide useful clinical information [27].

Both subjective and objective total sleep time values within our data are below the recommended amount of sleep for the age group, according to the American Academy of Sleep Medicine (10–13 h) [28]. Regarding subjective data, our finding contrasts with that reported by several studies, including Bruni et al., where a significant increase in sleep duration was found in all age groups except 1–3 years old. Moreover, even with Lokhandwala et al. findings, where actigraphic measurements were obtained from 16 preschool-aged children reporting an increase in total sleep time during the COVID-19 pandemic compared to pre-pandemic data (10.52 h vs. 10.48 h), also reporting earlier wake times association with greater negative coping expression [29]. 

In China, Liu et al. described longer nightly sleep durations compared to our sample and to a pre-COVID-19 assessment in their population, reporting a mean of 10.38 h of sleep time per night and 11.09 h of total 24 h sleep duration. In addition, they found a decrease in sleep disturbances prevalence, finding 55.6% of sleep disturbances during the COVID-19 pandemic compared to 77.7% in their previous sample. They related this finding to the influence of modifiable behavioral practices, more flexible daily schedules, reduced electronic device use, and a positive home environment [30].

Regarding the home environment, Lionetti et al. suggested that among other factors to assess as moderators in preschool children’s sleep during the COVID-19 pandemic, the influence of individual and environmental variables should be considered [31]. For example, within our sample, having the antecedent of being premature was found to be associated with higher sleep disturbance severity. This finding was previously observed by Brockmann et al., who reported that preterm children had higher sleep disturbance scores, had decreased sleep quality, and developed long-term sleep problems [32]. Furthermore, studies also reported shorter sleep duration and irregular sleep schedules in preterm children than in children born at term [33]. 

Although within our sample, children did not have any preexisting diagnosis of sleep disturbances or mental health disorders, the presence of those individual factors could have deteriorated sleep quality during the COVID-19 pandemic, as found in Mackenzie et al. study in Canada, where parents of children with preexisting insomnia, perceived and influence of maladaptive bedtime routines, stress and reduced exercise to their children’s worsened sleep [3]. 

On the other hand, some other relevant environmental factors that can be related to a higher risk of sleep disturbances in preschooler children during the COVID-19 pandemic are parental and familiar variables. Andersen et al. found an association between maternal anxiety, insomnia, and fear with the presence of sleep disorders in young children. Those sleep disturbances could, in turn, influence children’s behavioral difficulties that impact maternal well-being. Furthermore, the financial impact of the pandemic was related to mothers’ anxiety which was also associated with children’s sleep disturbances [13]. In that same line, Gupta et al. found that a family’s lower household income was related to increased sleep onset latency in infants and toddlers in the United States [34]. 

In Chile, Aguilar-Farias et al. found that more educated parents between 35 and 45 years old tended to restrict their children’s physical activity more and provide more opportunities for children to increase their screen time. This finding could be related to the need for mothers to balance carrying out work activities at home and entertaining their children while working [35]. The latter could be particularly true for our sample of parents, as it was composed predominantly of employed women, with a mean age of 36.8 years old and a college degree. 

Parents reported an increase in the use of electronic devices, where electronic tablets were the most used device. Furthermore, according to actigraphic data, its use in the bedroom near bedtime was associated with the presentation and severity of sleep disturbances, total sleep time, and motionless sleep time reduction. Although actigraphy does not determine sleep stages as a polysomnographic register would, the registered decrease in motionless sleep could signal an influence in deep sleep, where crucial neurophysiological phenomena such as glucose metabolism, hormone release, cognition, and immunity processes occur [36]. Moreover, this finding agrees with that reported by Ju-Kim et al., who documented that children’s sleep disturbances were associated with both tablet and smartphone use during the COVID-19 pandemic [37]. 

Even though several studies have already reported on the link between the use of conventional electronic devices and sleep disturbances, special considerations should be focused on the impact of electronic tablets on preschool children’s sleep. Their ease of handling due to a touchscreen, its portability that allows constant use [38], and its possibility of being a frequent source of light exposure that theoretically could decrease melatonin release [39] can imply a significant clinical impact. In a pre-pandemic study by Chindamo et al., the daily use of a tablet or smartphone increased the probability of a shorter total sleep time and longer sleep latency, regardless of other factors, such as temperament and screen exposure from conventional media (e.g., TV and video games) [40]. As was also discussed by Lan et al., using portable electronic devices has been associated with sleep disturbances and short sleep duration. At the same time, electronic devices in the bedroom and their use at bedtime can increase the risk of a discrepancy between the circadian rhythm and social time (social jetlag) in preschool children [41]. The latter gets even more relevant as social jet lag has been associated with behavioral problems in preschool children [42]. A relationship between the presence of screens in bedrooms and evening screen exposure with more sleep disturbances and higher behavioral and hyperactivity symptoms has also been described by Cavally et al., concluding that the recommendation to avoid screens in children’s bedrooms and limiting evening exposure to them should be encouraged [43]. 

Within the analyzed data, higher sleep disturbance severity was associated with externalizing and internalizing mental health symptoms, such as behavioral difficulties and emotional distress. Cellini et al. reported similar findings, indicating a marked delay in children’s sleep time during prolonged confinement and a slight deterioration in sleep quality. This was also associated with increased emotional, behavioral, and hyperactive symptoms, which were also related to their mothers’ emotional difficulties [44]. In our study, 15.6% of the children reported mental health deterioration symptoms without any previous diagnoses. This finding can relate to Altena et al.´s proposal that isolation during the COVID-19 pandemic could have compromised children’s ability to regulate behavior and emotions and consequently exposed them to sleep problems and vice versa [45]. 

According to the evidence, sleep and mental well-being differences during the pandemic confinement appear to be related to the specific developmental stage. In Australia, a study by Stone and colleagues on an adolescent population demonstrated multiple beneficial changes in mood and sleep associated with homeschooling during the COVID-19 pandemic. They found that during remote learning, adolescents slept more in line with their endogenous circadian rhythms, positively affecting sleep duration, augmenting 22 min of sleep reported by actigraphy. A reduction in their anxiety scores and decreased perceived stress was also noted (even though it was not mediated by sleep) [46]. Bruni et al. also discussed a similar finding where adolescents only reported difficulties falling asleep in their observations, being the group with the most significant bedtime delay, without describing any other sleep alteration [25]. Gruber et al. reported the disappearance of social jet lag for adolescents during the COVID-19 pandemic, with longer sleep durations and less daytime sleepiness [47]. 

The findings of our study should be interpreted considering certain limitations. First, the cross-sectional study design represents a limited period of assessment, which limits the possibility of drawing stronger conclusions and establishing causality. Second, the use of WhatsApp groups could contribute to a selection bias by not including parents who are not present in them. Third, the open invitation to participate might have drawn parents concerned about their children’s sleep and mental well-being, influencing the reported symptoms. Finally, the small sample size may represent a limitation for the overall generalization of the study findings.

Further research that focuses on the behavioral and biological influences of portable electronic devices, such as tablets, on sleep in early childhood could support this study’s findings and provide more specific recommendations and educational strategies regarding the use of these electronic devices in that stage of development. Likewise, obtaining post-confinement sleep measures due to the COVID-19 pandemic can provide more information about the current sleep status of children and their routine modifications. 

## 5. Conclusions

After a year of confinement due to the COVID-19 pandemic, sleep disturbances were highly prevalent in preschool-aged children, and their total sleep time duration was significantly impacted by tablet use near bedtime. Because healthy sleep is essential for optimal development among children and is intimately associated with their mental well-being, further exploring the consequences of these findings and their post-confinement evolution is necessary. This exploration can influence the design of age-tailored interventions to prevent specific risk factors and protect preschool children’s overall health. 

## Figures and Tables

**Figure 1 ijerph-20-04386-f001:**
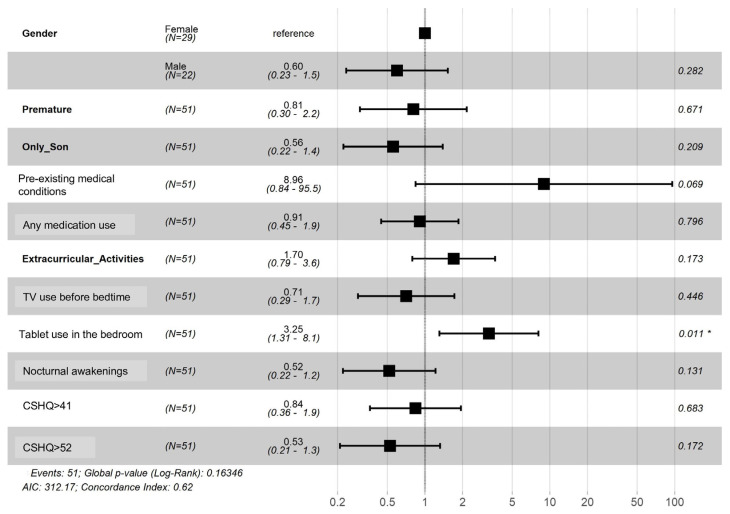
Hazard Risk Ratio for All-Week Total Sleep Time Decrease. * Statistically significant value (*p* 0.011).

**Figure 2 ijerph-20-04386-f002:**
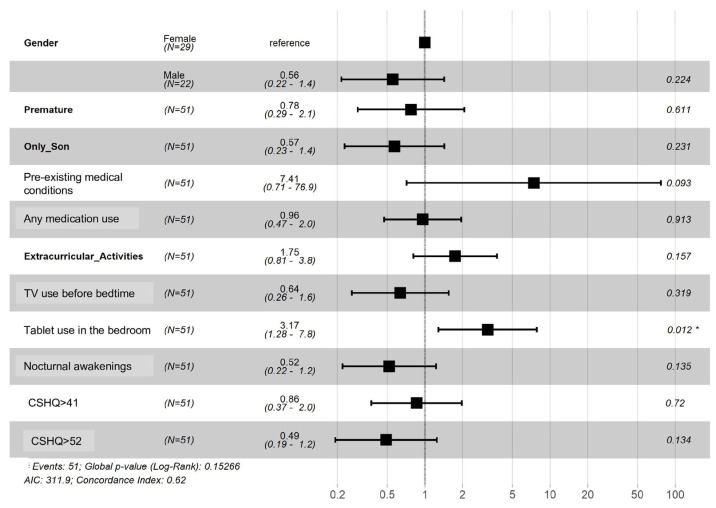
Hazard Risk Ratio for All-Week Motionless Sleep Time Decrease. * Statistically significant value (*p* 0.012).

**Table 1 ijerph-20-04386-t001:** Children’s Demographics and Electronics Use During the Day.

Demographic Variable		CSHQ < 41	CSHQ > 41	*p*	CSHQ > 52	*p*	OR [CI 95%]
		N 16, 31.37%	N 23, 45.1%		N 12, 23.53%		
Gender	F	8, 27.59%	12, 41.38%	0.55	9, 31.03%	0.19	2.85 [0.67–12.15]
	M	8, 27.59%	11, 37.93%	0.55	3, 10.34%	0.19	0.35 [0.08–1.5]
Prematurity		1, 3.45%	2, 6.9%	0.41	4, 13.79%	0.04	6 [1.12–32.24]
Only son		3, 10.34%	3, 10.34%	0.99	3, 10.34%	0.99	1.83 [0.38–8.81]
Any medication use		6, 20.69%	9, 31.03%	0.76	6, 20.69%	0.51	1.6 [0.43–5.89]
Extracurricular activities		6, 20.69%	8, 27.59%	0.99	5, 17.24%	0.74	1.28 [0.34–4.78]
Age (mean ± SD)		5.13 ± 1.45	5.61 ± 1.27	0.57	4.83 ± 0.83	0.16	
Number of siblings (median, IQR)		1 [1~3]	2 [1~2]	0.36	1 [1~2]	0.19	
TV hours per day (median, IQR)		2.5 [1~3]	3 [2~4]	0.17	2 [1.5~2.5]	0.32	
Computer hours/day (median, IQR)		0 [0~1]	0 [0~4]	0.83	0 [0~0]	0.13	
Tablet hours/day (median, IQR)		3.5 [1~6]	3 [2~5]	0.89	4 [2.5~7]	0.27	
Smartphone hours/day (median, IQR)		0 [0~1]	0 [0~2]	0.34	0.5 [0~1]	0.71	

**Table 2 ijerph-20-04386-t002:** Sleep Variables.

Sleep Variable	CSHQ < 41	CSHQ > 41	*p*	CSHQ > 52	*p*
	N 16, 31.37%	N 23, 45.1%		N 12, 23.53%	
Children´s Sleep Habits Questionnaire (median IQR)					
Bedtime resistance	6.5 [6~8.5]	10 [8~11]	<0.01	12.5 [11.5~15]	<0.01
Sleep onset delay	1 [1~1]	1 [1~1]	0.02	2 [1~3]	<0.01
Sleep duration	3 [3~3]	3 [3~4]	<0.01	4 [3.5~5]	<0.01
Sleep anxiety	2 [2~2]	2 [2~3]	0.01	3.5 [2.5~4]	<0.01
Night waking’s	3 [3~4]	5 [3~5]	<0.01	6 [4.5~8.5]	<0.01
Parasomnias	8 [7.5~8]	9 [8~9]	<0.01	9 [8~12]	<0.01
Sleep disordered breathing	3 [3~3]	3 [3~3]	<0.19	3 [3~3]	0.81
Daytime sleepiness	10 [8~10]	13 [11~15]	<0.01	16 [12.5~17.5]	<0.01
Sleep diary (median, IQR)					
Total sleep time (hours)	10.31 [9.79~10.94]	9.61 [9.17~10.44]	0.03	9.55 [8.9~9.96]	0.30
Sleep latency (minutes)	11.93 [8.43~15.43]	17.14 [7.71~21.43]	0.08	13.93 [8.93~28.2]	0.36
Sleep efficiency	98.23 [97.55~98.5]	97.06 [96.45~98.5]	0.04	97.44 [95.5~98.4]	0.33
Actigraphy (mean ± SD, median, IQR)					
Total sleep time (hours)	7.46 ± 3.24	6.81 ± 1.1	0.27	6.77 ± 1.2	0.72
Total awake time (hours)	16.54 ± 3.24	17.19 ± 1.1	0.27	17.23 ± 1.2	0.72
Motionless sleep	7.4 ± 3.26	6.78 ± 1.11	0.29	6.73 ± 1.19	0.74
Light activity during sleep	0.18 [0.07~0.46]	0.09 [0.07~0.24]	0.22	0.12 [0.04~0.18]	0.58
Moderate activity during sleep	0.02 [0.02~0.06]	0.02 [0.02~0.05]	0.66	0.03 [0.02~0.06]	0.19
Nocturnal awakenings	4 [3.21~4.86]	3.57 [3~4.07]	0.93	4 [3.43~4.71]	0.18

**Table 3 ijerph-20-04386-t003:** Use of Electronics Devices Before Bedtime.

Electronic Used before Bedtime	CSHQ < 41	CSHQ > 41	*p*	CSHQ > 52	*p*	OR [CI 95%]
	N 16, 31.37%	N 23, 45.1%		N 12, 23.53%		
TV in the bedroom	10, 34.48%	15, 51.72%	0.75	9, 31.03%	0.72	1.68 [0.39–7.24]
Watching TV before sleep	13, 44.83%	20, 68.97%	0.99	8, 27.59%	0.21	0.36 [0.08–1.6]
Tablet in the bedroom	2, 6.9%	7, 24.14%	0.05	8, 27.59%	0.01	6.67 [1.62–27.38]

**Table 4 ijerph-20-04386-t004:** Children’s Sleep Disturbances and Mental Well-Being.

SDQ Variables	CHSQ > 41	*p*	CHSQ >52	*p*
	N 23, 45.1%		N 12, 23.53%	
SDQ Emotional Distress Score	0 [0~1]	0.14	2.5 [0~5]	<0.01
SDQ Behavioral Difficulties Score	1 [1~2]	0.12	3 [2~3.5]	<0.01
SDQ Hyperactivity Score	5 [2~6]	0.87	6.5 [5~7.5]	0.12
SDQ Difficulties Getting Along with Other Children Score	1 [0~1]	0.98	1 [0~2.5]	0.39
SDQ Prosocial Behavior Score	9 [8~10]	0.85	8 [8~9]	0.22
SDQ Total Score	7 [6~11]	0.23	13 [9.5~ 16]	<0.01
SDQ Externalizing Score	6 [4~8]	0.43	10 [8~11]	0.01
SDQ Internalizing Score	1 [0~3]	0.29	6 [1~6.5]	<0.01

## Data Availability

Data are available upon reasonable request through the corresponding author Julieta Rodriguez de Ita; julyrdz@tec.mx.

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
