# Peer review of "Sleep Disturbances and Mental Well-Being of Preschool Children during the COVID-19 Pandemic in Mexico"

_ijerph, 2023, doi:10.3390/ijerph20054386_

Round 1
Reviewer 1 Report
page 8-line 240-241, is difficult to understand. Are the authors trying to say that the use of tablets close to bedtime resulted in a lack of motionless sleep that could signal an effect on deep sleep, which in turn affects neurophysiological phenomena?
If this is the case, I suggest rewording.
Mean/average and standard deviation is conventionally represented as mean+/- SD in medical literature. Std is used more in engineering. (Table 1)
The authors mention that 43.2% of children were in the third year of kindergarten. Did the years in school make a difference to the sleep patterns? Did children in the first year of kindergarten have less problems than the ones in the third year?
Author Response
We are very grateful, and thank you for your kind comments that have enriched our paper. Please see the attachment.

Reviewer 2 Report
This study, based on a small sample of children, analyses the impact of confinement on the sleep of young children. There are many studies on this issue, but the authors do not take into account and therefore do not analyse the key factors of this problem. The only originality of the study is the objective measurement by actimetry. It should be coupled with a post-covid measurement of the same children to be particularly relevant. As such all the introduction and conclusion should be rewritten to reflect the current literature, otherwise a "short communication" format would be sufficient to describe the actimetry data that supports the current literature.
1. Introduction :
Major concerns : I advise to rewrite the introduction by orienting the argument on the one hand to a comprehensive literature review and on the other hand to focus on the need for confirmation of previous observations by objective measurements.
“Nowadays, there are some limitations in pre-pandemic and pandemic existing sleep 59 studies, such as the underrepresentation of preschool children's sleep data.”
Author do not know the literature :
• Lecuelle, F., Leslie, W., Huguelet, S., Franco, P., & Putois, B. (2020). Did the COVID-19 lockdown really have no impact on young children's sleep? Journal of Clinical Sleep Medecine, 16(12):2121. (IF-2019: 3.6)
• Anders R., Lecuelle F., Perrin C., Ruyter S., Franco P., Huguelet S., Putois, B. (2021). The Interaction between Lockdown-Specific Conditions and Family-Specific Variables Explains the Presence of Child Insomnia during COVID-19: A Key Response to the Current Debate. International Journal of Environmental Research and Public Health (IF-2020: 3.39).
2. Materials and Method
How were the participants recruited? Is there not a selection bias?
4. Discussion
Major Concern :
Discuss the prevalence results and key factors in relation to other articles on the subject.
Minor Concern :
· Line 248-262 : Linking screen exposure in the evening to sleep and daytime behaviour:
Cavalli, E., Royce, A., Chaussoy, L., Herbillon, V., Franco, P., Putois, B. (2021). Screen exposure exacerbates ADHD symptoms indirectly through increased sleep disturbance. Sleep Medicine, 83(4). (IF-2019: 3.6)
· “Regarding the home environment, parents reported an increase in electronic devices 236 use (not related to schooling),”
· What is the link between increased screen use in relation to confinement and not schooling?
· “In Australia, a study on an adolescent” Please, write the reference of the study.
5. Conclusion
I advise to focus only on the main contribution of this study which is the objective measurement
6. References Recommendation
Camacho-Montaño LR, Iranzo A, Martínez-Piédrola RM, Camacho-Montaño LM, Huertas-Hoyas E, Serrada-Tejeda S, García-Bravo C, de Heredia-Torres MP. Effects of COVID-19 home confinement on sleep in children: A systematic review. Sleep Med Rev. 2022 Apr;62:101596. doi: 10.1016/j.smrv.2022.101596. Epub 2022 Feb 3. PMID: 35183816; PMCID: PMC8810276.
MacKenzie NE, Keys E, Hall WA, Gruber R, Smith IM, Constantin E, Godbout R, Stremler R, Reid GJ, Hanlon-Dearman A, Brown CA, Shea S, Weiss SK, Ipsiroglu O, Witmans M, Chambers CT, Andreou P, Begum E, Corkum P. Children's Sleep During COVID-19: How Sleep Influences Surviving and Thriving in Families. J Pediatr Psychol. 2021 Sep 27;46(9):1051-1062. doi: 10.1093/jpepsy/jsab075. PMID: 34472600; PMCID: PMC8522399.
Lionetti F, Fasolo M, Dellagiulia A. On the role of moderators on children's sleep health in response to COVID-19. J Clin Sleep Med. 2021 Feb 1;17(2):353-354. doi: 10.5664/jcsm.8948. PMID: 33118926; PMCID: PMC7853235.
Okely AD, Kariippanon KE, Guan H, Taylor EK, Suesse T, Cross PL, Chong KH, Suherman A, Turab A, Staiano AE, Ha AS, El Hamdouchi A, Baig A, Poh BK, Del Pozo-Cruz B, Chan CHS, Nyström CD, Koh D, Webster EK, Lubree H, Tang HK, Baddou I, Del Pozo-Cruz J, Wong JE, Sultoni K, Nacher M, Löf M, Cui M, Hossain MS, Chathurangana PWP, Kand U, Wickramasinghe VPP, Calleia R, Ferdous S, Van Kim T, Wang X, Draper CE. Global effect of COVID-19 pandemic on physical activity, sedentary behaviour and sleep among 3- to 5-year-old children: a longitudinal study of 14 countries. BMC Public Health. 2021 May 17;21(1):940. doi: 10.1186/s12889-021-10852-3. PMID: 34001086; PMCID: PMC8128084.
Wearick-Silva LE, Richter SA, Viola TW, Nunes ML; COVID-19 Sleep Research Group. Sleep quality among parents and their children during COVID-19 pandemic. J Pediatr (Rio J). 2022 May-Jun;98(3):248-255. doi: 10.1016/j.jped.2021.07.002. Epub 2021 Aug 23. PMID: 34480854; PMCID: PMC8432904.
Author Response
We are very grateful for all your precise comments. Please see the attachment for the response.

Round 2
Reviewer 2 Report
Thank you for this new version. You have considerably improved the manuscript by your thorough analyse of the literature on the subject. I just suggest a final check of the English by a native speaker.
Best regards,